# Multigrid Distributed Deep CNNs for Structural Topology Optimization

**Jaydeep Rade**[1] **Aditya Balu**[1] **Ethan Herron**[1] **Anushrut Jignasu**[1] **Sergio Botelho**[2] **Santi Adavani**[2]
**Soumik Sarkar**[1] **Baskar Ganapathysubramanian**[1] **Adarsh Krishnamurthy**[1]

Iowa State University [1]
RocketML Inc. [2]
adarsh@iastate.edu[1], santi@rocketml.net[2]

## Abstract

Structural topology optimization with traditional approaches is compute-intensive, mainly due to multiple finite element analysis iterations required to evaluate the component's performance during the optimization process. This computation cost scales up when performed on 3D high-resolution geometries. Researchers have developed deep learning (DL) based approaches, but these methods were demonstrated mainly using low-resolution 3D geometries (with a typical resolution of $32 \times 32 \times 32$). We propose a DL-based method trained with a convolutional neural network (CNN) on high-resolution 3D geometries $128 \times 128 \times 128$. With the initial strain energy (objective function of structural topology optimization) and target volume fraction (% material to be preserved after optimization) as the only inputs to the CNN, we predict the final optimized topology while maintaining the volume fraction constraint. To train the CNN at a high resolution is again a computational challenge. Therefore, we propose multi-resolution CNN, where we train the network at a lower resolution and then transfer the learned network to continue training at a higher resolution. Further, we significantly speed up the training time by $4.77\times$ using distributed deep learning framework on GPU clusters (PSC Bridges-2).

## Introduction

Topology optimization has been used for designing components with optimal performance (Orme et al. 2017; Liu and Ma 2016). Structural topology optimization, initially developed by Bendsøe and Kikuchi (1988), is a set of numerical design optimization methods to find the optimal distribution of the material in the initial design domain to generate designs with optimal performance while removing the material to satisfy a volume fraction constraint. The main challenge in performing structural topology optimization at high resolution is that it is computationally intensive, mainly because of evaluating the objective function at each iteration. The objective function is computed using numerical solution approaches such as finite element analysis, which are computationally expensive. Performing topology optimization with this computational challenge, especially for high-resolution geometries, could take a few hours to days. The natural solution for this problem is to use deep learning methods to

perform topology optimization (Rade et al. 2021; Chi et al. 2021; Lagaros, Kallioras, and Kazakis 2020) and reduce numerical simulations.

However, training deep learning models for high-resolution 3D geometries is also compute-intensive. Hence, we implement a training algorithm similar to the multigrid approach (Balu et al. 2021). Here we first train the deep convolutional neural networks (CNNs) on low-resolution 3D geometries since the training on low-resolution geometries is faster than high-resolution geometries. Then using the learned weights of CNN, we continue training further on high-resolution 3D geometries. In addition to using the multigrid approach, we incorporate a data-parallel distributed training scheme for deep CNN on the GPU cluster. In a data-parallel distributed deep learning scheme, multiple copies of the model are trained simultaneously to optimize a single objective function, which is used to overcome the memory limitation on a single device.

To summarize our contributions, we propose a deep learning-based topology optimization framework in 3D and explore methods to address the computational challenges at large resolutions in a 3D voxel grid. To this end, our specific contributions are:

1. We use a "multigrid" inspired approach to develop a transfer learning framework to progressively train networks for training at very large resolution voxel grids for predicting the optimal shape from topology optimization.

2. We use a data-parallel distributed deep learning framework to accelerate the training process in each resolution.

3. Using the multigrid approach and the data-parallel distributed learning approach, we have been able to accelerate training time by $4.77\times$ while training at the voxel resolution of $128 \times 128 \times 128$.

We now begin by covering some preliminaries for our framework. Next, we cover the details of the methods proposed and then show some preliminary results of our framework and provide some concluding remarks.

## Formulation and Preliminaries

In this section, we will explain the mathematical formulation for structural topology optimization and the idea of the multigrid training scheme and data-parallel distributed deep learning.

## Topology Optimization

Structural topology optimization is a minimization problem represented as:

$$\text{Minimize: } C(\rho, \mathbf{U}) = \mathbf{U}^T \mathbf{K} \mathbf{U} = \sum_{e=1}^{N} \rho_e^p u_e^T k_0 u_e$$

$$\text{subject to: } \mathbf{K}(\rho)\mathbf{U} = \mathbf{F} \tag{1}$$

$$g_i(\rho) = \frac{V(\rho)}{V_0} - V_f \leq 0$$

$$0 < \rho \leq 1$$

where $\mathbf{U}$ and $\mathbf{F}$ are the global displacement and force vectors, respectively, K is the global stiffness matrix, and $u_e$ and $k_e$ are the element displacement vector and stiffness matrix, respectively. $C(\rho, \mathbf{U})$ represents the objective function of structural topology optimization, which is the total strain energy of the system and $\rho$ is the density of an element. The volume fraction constraint is given by $g_i$, where $V(\rho)$ is the volume of the design at any iteration of the problem and $V_0$ is the initial volume of the design, and $V_f$ is the target volume fraction. This ensures that the target volume fraction is maintained throughout the optimization process. Structural topology optimization using the solid isotropic material with penalization (called SIMP (Bendsøe 1989)) algorithm, where the stiffness for each element is described as, $E = E_{min} + \rho^p (E_{max} - E_{min})$. Here, $p$ is the parameter used to penalize the element density close to 1.0. A typical SIMP-based topology optimization pipeline is shown in Algorithm 1. More methods for structural topology optimization such as level-set methods (Wang, Wang, and Guo 2003) and evolutionary optimization methods (Das, Jones, and Xie 2011; Xie and Steven 1993) are also popularly employed. All these methods have the common challenge of performing several iterations of finite element analysis, making it computationally challenging.

---

Algorithm 1: SIMP topology optimization (Bendsøe 1989)

---
**Input** : $\mathcal{S}$, L, BC, $V_0$
**Output** : $D_{fin}$(set of all densities for each element, $\rho$)
Load design; apply loads and boundary conditions
  Initialize: $D_0 \rightarrow V_0/\int_\Omega d\Omega$
  Initialize : $ch = \inf$
  **while** $ch < threshold$ **do**
  | Assemble global stiffness matrix $\mathbf{K}$ for element stiffness matrix $k_e(\rho_e)$
  | Solve for $\mathbf{U}$, using $\mathbf{K}$, loads (L) and boundary conditions (BC)
  | Compute objective function,
  | $\mathbf{C} = \mathbf{U}^T \mathbf{K} \mathbf{U} = \sum_{e=1}^{N} \rho^p u^T k_e u$
  | Perform sensitivity analysis, $\frac{\partial c}{\partial e} = -p\rho^{(p-1)} u^T k_e u$
  | Update the densities ($D_i$) using a optimality criterion
  | $ch = ||D_i - D_{i-1}||$

---

## Multigrid Approach

The multigrid methods (MG) in numerical analysis are used for solving the partial differential equation (PDEs) on a hierarchy of meshes and sequentially projecting and solving the problem on these meshes. The main idea of MG is to project a fine grid problem onto a coarser grid to convert low-frequency errors to high-frequency errors, which can be smoothened through multiple iterations. This technique can be extended to CNNs as the weights of CNNs are independent of the spatial resolution of its input (Ke, Maire, and Yu 2017; Balu et al. 2021). The idea exploited here for the multigrid training is that the weights of one resolution can be used for another resolution without any change in the structure due to the CNN weights being independent of each other when extending from one resolution to another. This approach, in effect, helps us learn the features better as we go from one resolution to another.

While several strategies to perform multigrid have been explored in the context of 3D deep learning (Balu et al. 2021), half-V seems to be the most trivial and effective strategy for a learning problem. In the half-V multigrid strategy, we start from the coarsest grid from the hierarchy of meshes and train for a large number of epochs till convergence. Using the trained weights, we can now fine-tune the weights with fewer epochs in the finer level grid of the hierarchy of meshes. Finally, you perform very few (¡ 5 epochs) in the final fine resolution to train the network completely. While the idea of multigrid was originally meant for solving PDEs, this strategy for progressively training deep CNNs from the lower resolutions to higher resolutions seems to be extendable to other applications such as this work on multigrid training for Topology Optimization. Therefore, we use this strategy with three hierarchy levels for training, beginning with $32 \times 32 \times 32$ to $128 \times 128 \times 128$.

## Distributed Deep Learning

In the era of big data, it is hard to fit the data and sometimes larger deep learning models in the memory of a single machine (either CPU or GPU). So it becomes necessary to use multiple machines to train the deep learning models on the big dataset. There are mainly two types of parallelism: (i) Data parallelism and (ii) Model parallelism. In data parallelism, the whole data is divided over multiple machines; on the other hand, in model parallelism, the model layers are distributed across multiple machines. In this paper, we take advantage of data-parallel distributed deep learning to accelerate the training process.

To train the distributed deep learning model using the data-parallel technique, identical model copies are copied to the independent devices and trained simultaneously. Each device then works on different sets of data, and the devices collectively update the model (Chen, Yang, and Cheng 2019). The training data is divided equally among the devices and is further divided into local mini-batches, which are asynchronously processed through the forward and backward pass (Balu et al. 2021). Each device calculates the gradients locally, and then the global gradients are computed by averaging the local gradients using an `all-reduce` operation.

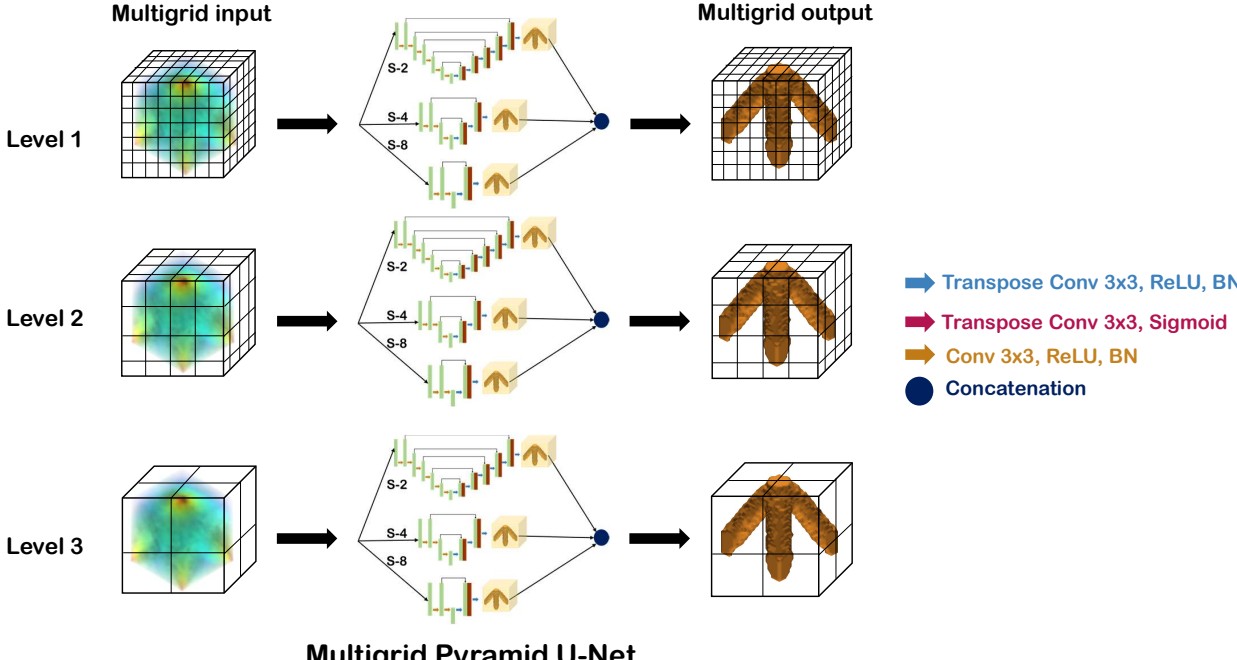

**Figure 1**: Pyramid U-Net is the hybrid version of U-Net (Ronneberger, Fischer, and Brox 2015; Çiçek et al. 2016) and PSPNet (Zhao et al. 2017). We leverage the idea of PSPNet to U-Net architecture as a module and implement it at multiple steps which has different strides of convolution at each step. Inputs to the network are strain energy of initial geometry and target volume fraction and the output is final optimized geometry shape.

Once the global gradient is computed, it is broadcasted to all the devices to update the local model parameters. As all devices update the local model parameters using the same global gradients, the model parameters are consistent through all the devices.

## CNN Architecture: Pyramid U-Net

To learn the non-trivial transformation from the initial shape to the optimal geometry shape in the structural topology optimization process, we propose Pyramid U-Net, a fully 3D convolutional neural network. Pyramid U-Net is a hybrid of U-Net (Ronneberger, Fischer, and Brox 2015; Çiçek et al. 2016) and pyramid scene parsing network (PSPNet) (Zhao et al. 2017) architectures. The Pyramid U-Net architecture comprises multiple U-Net modules (each working on the grid with different strides, e.g., first module has a stride of 2, the second has a stride of 4, and thirst with a stride of 8). This helps in capturing the features at different scales. Each U-Net module in the network is an encoder-decoder style architecture with skip connections from an encoder to the decoder. These skip-connections enable strong localization during up-sampling (or decoding) using the contextual information from downsampling (or encoding) layers. We implement the U-Net architecture at three levels: Stride-2, Stride-4, and Stride-8. Each level has different strides (2, 4, and 8, respectively) for pooling operation along the downsampling and upsampling path. This idea comes from PSPNet, where the global prior is computed at different scales to enrich the context information extraction from the input.

The input to the network is concatenated tensor of initial strain energy and target volume fraction. We feed this input tensor to the network and obtain the final optimal density. The dimensionality of the strain energy, the volume fraction, and the optimal density are all the same as the mesh size (the voxel grid size in this case).

## 3D Dataset

The 3D data used is generated using ANSYS Mechanical APDL v19.2, which uses the SIMP method for structural topology optimization. As an initial geometry, we use a cube of a length of 1 meter. The mesh of this cube contains 31093 nodes and 154,677 elements, and each element consists of 8 nodes. We use a diverse set of boundary and loading conditions available in ANSYS software such as Nodal Force, Surface Force, Remote Force, Pressure, Moment, Displacement. This ensures that the dataset has a variety of samples from the set of complete distributions of topologies originating from the cube. The obtained geometries are in mesh format; we convert them into voxel format to train CNNs. For the multi-resolution purpose, voxelization is performed at three resolutions: $32^3$, $64^3$, and $128^3$. We generated a total of 60,000 (60K) samples. We split the dataset into training and testing sets with 50K and 10K samples, respectively. Further, we build five sets of the training data, each having sizes of 10K, 20K, 30K, 40K, and 50K.

Table 1: Comparison between multigrid and single resolution CNN. We compare the number of samples for training, the number of epochs trained, the BCE loss value on the test data set, and the training time.

| Method | # Training samples | # Epochs | BCE loss | Time |
|---|---|---|---|---|
| Single resolution ($128^3$) | 50K | 25 | 0.1658 | 23hrs 36min |
| Multigrid ($32^3/64^3/128^3$) | 50K/30K/30K | 15 (5 each) | 0.1735 | 4hrs 57min |

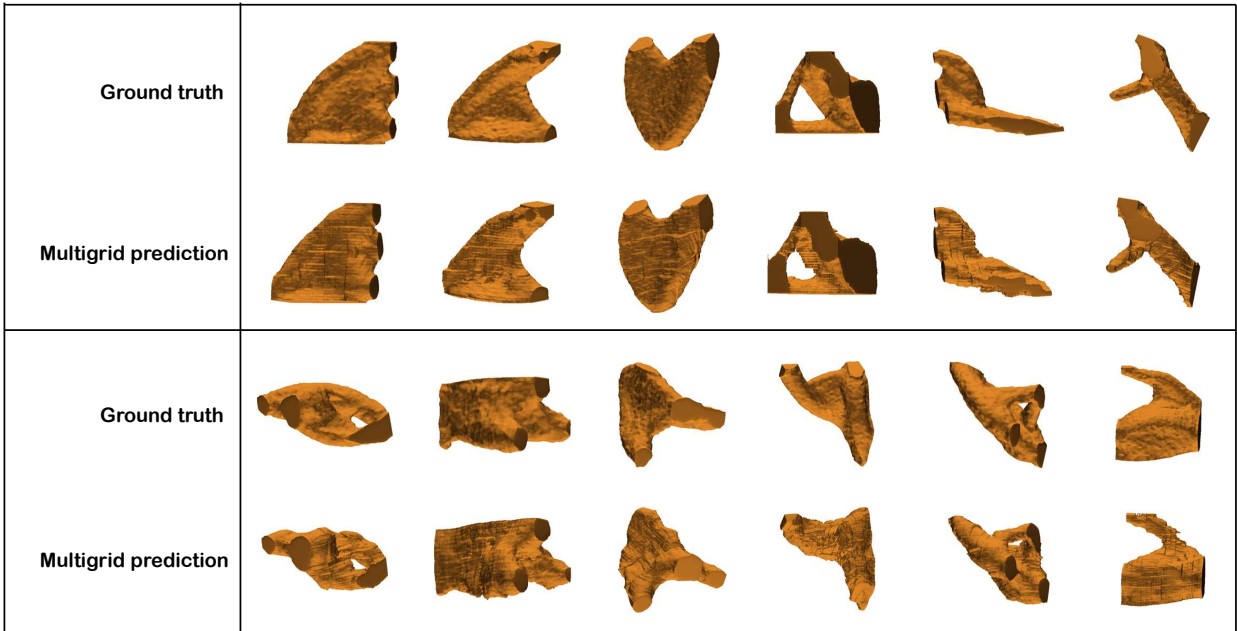

Figure 2: Comparison of geometry shapes obtained with SIMP and the multigrid approach. The ground truth here is the output of the SIMP method.

## Results and Discussion

The total number of parameters for CNN we used is 76,153,604 (76 Million). We use the SGD-based Adam (Kingma and Ba 2014) optimizer for training, with a learning rate of 0.0003 and batch size of 64, so the minimum local batch size is 2 per GPU. We use the binary cross-entropy (BCE) function to guide the optimizer to calculate the loss between the predicted and the target geometry. We performed the neural network training on the Bridges-2 GPU cluster, which contains 8 NVIDIA Tesla V100 GPUs, each with 32GB memory per device per node. The training was performed on as many as 4 nodes (32 GPUs) using 8 devices per node.

We train the CNN with the coarser geometry resolution of $32^3$ several epochs. Next, we transfer the learning to the next finer resolution of $64^3$ by initializing the training with the weights saved after training on the previous coarser resolution. Similarly, we trained further for several epochs and then used these weights as initialization for the next finer resolution of $128^3$ and trained for more epochs.

In addition, to motivate multigrid training, we also performed training using the only single resolution of $128^3$ instead of using multi-resolution geometries and compared the timing and performance with the multigrid training approach. We trained the CNN using single resolution with the same hyper-parameters as used for multigrid training.

From Table 1, we observe that, with the less number of epochs (15 epochs) trained using multigrid approach yields comparable loss value compared when trained using single resolution training with more number of epochs (25 epochs). In multigrid training, we trained CNN for 5 epochs using each resolution level, resulting in 15 epochs of training. Additionally, we reduced the number of samples from 50K to 30K in multigrid training where we transferred the learning from $32^3$ resolution to finer resolution; on the other hand, we used 50K samples for single training resolution CNN. So we see the advantage of the multigrid approach where we can train for fewer epochs and even train on fewer training samples at a finer resolution level, we get a comparable loss value with a speedup of $4.77\times$.

Further, we evaluate the multigrid approach with the SIMP method by visualizing the final geometry obtained by both methods using the marching cubes algorithm. In Figure 2, we compare the ground truth obtained through the SIMP method and the multigrid prediction. From Figure 2, we observe that the multigrid approach can predict the shape of the final geometry much accurately.

## Conclusions

In this paper, we leveraged the idea of the multigrid approach to accelerate the learning by performing the topology optimization at a hierarchy of mesh resolutions. Further to speed up the training, we implemented distributed deep learning approach where we trained the CNN on multiple GPUs on a GPU cluster. With the combined advantage of the multigrid approach and distributed deep learning, we trained the CNN significantly faster with a high-resolution $128 \times 128 \times 128$ geometries. For future work, we want to extend this idea to the megavoxel domain with a resolution as fine as $512 \times 512 \times 512$ and also perform deep learning-based topology optimization for more complicated and real-world structures.

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
