# OpenReview forum: "Multigrid Distributed Deep CNNs for Structural Topology Optimization"
_AAAI.org/2022/Workshop/ADAM — AAAI 2022 Workshop ADAM_

### Official Review · Reviewer_Pt3g · 2021-11-30
**Review for Multigrid Distributed Deep CNNs for Structural Topology Optimization**

**Rating:** 7
**Confidence:** 2

**Review:**

The paper mainly proposed a convolutional neural network (CNN) that can cope with high-resolution 3D geometry. Transfer learning is used to training CNN from low resolution to high resolution.  Due to the high computational cost on training CNNs on high-resolution 3D geometries, authors craftly utilize multigrid approach to reduce the training time for the deep learning framework. Although the multigrid approach originated from resolving patrial differential equations, the paper creatively explored extendable applications on training CNNs and data parallel technique to alleviate the workload of machines.

This multigrid distributed CNNs accelerate the training process by 4.77x, which achieved comparable loss value with single resolution method. The multigrid predicted geometries are accurate when comparing to the solid isotropic material with Penalization method (SIMP). For the clarity it would be useful if authors demonstrate the multigrid approach with some schematic diagram of the algorithm. Exploring more generic extendable relationship between various type of data would be promising future goals.

---

### Official Review · Reviewer_w87H · 2021-11-30
**Review of Multigrid Distributed Deep CNNs for Structural Topology Optimization**

**Rating:** 7
**Confidence:** 5

**Review:**

This paper has combined ideas from multi-grid approach, transfer learning of CNN weights across spatial resolutions, and data-parallel distributed training to build a CNN framework for performing topology optimization at higher resolution compared to previous works. The authors have demonstrated a 4.77x speed up with their approach at a high resolution of 128x128x128.

Pros: Well written paper, clever usage of Pyramid U-net, transfer learning within multi-grid approach

Cons: It is not clear what is the sensitivity of the prescribed multi-grid stride level discretization to the minimal allowable feature sizes on the optimized structure. Also it is not mentioned how boundary conditions are incorporated as inputs into the training, which eventually raises the question of how scalable this approach would be to different topology optimization problem with a wide spectrum of requirements (Volume fraction, boundary conditions, loading, multi-material etc.). It would be helpful to lay out the major technical, data and implementation challenges for transitioning this tool into a structural designer's workflow.